# New Species Can Broaden Myelin Research: Suitability of Little Skate, *Leucoraja erinacea*

**DOI:** 10.3390/life11020136

**Published:** 2021-02-11

**Authors:** Wiebke Möbius, Sophie Hümmert, Torben Ruhwedel, Alan Kuzirian, Robert Gould

**Affiliations:** 1Electron Microscopy Core Unit, Department of Neurogenetics, Max-Planck-Institute of Experimental Medicine, 37075 Göttingen, Germany; moebius@em.mpg.de (W.M.); sophie.huemmert@stud.uni-goettingen.de (S.H.); ruhwedel@em.mpg.de (T.R.); 2Cluster of Excellence Multiscale Bioimaging: from Molecular Machines to Networks of Excitable Cells (MBExC), University of Göttingen, 37073 Göttingen, Germany; 3Eugene Bell Center for Regenerative Biology and Tissue Engineering, Marine Biological Laboratory, Woods Hole, MA 02540, USA; akuzirian@mbl.edu; 4Whitman Science Center, Marin Biological Laboratory, Woods Hole, MA 02540, USA

**Keywords:** myelin evolution, little skate, oligodendrocytes, Schwann cells, elasmobranch, spinal cord, optic nerve, electron microscopy

## Abstract

Although myelinated nervous systems are shared among 60,000 jawed vertebrates, studies aimed at understanding myelination have focused more and more on mice and zebrafish. To obtain a broader understanding of the myelination process, we examined the little skate, *Leucoraja erinacea*. The reasons behind initiating studies at this time include: the desire to study a species belonging to an out group of other jawed vertebrates; using a species with embryos accessible throughout development; the availability of genome sequences; and the likelihood that mammalian antibodies recognize homologs in the chosen species. We report that the morphological features of myelination in a skate hatchling, a stage that supports complex behavioral repertoires needed for survival, are highly similar in terms of: appearances of myelinating oligodendrocytes (CNS) and Schwann cells (PNS); the way their levels of myelination conform to axon caliber; and their identity in terms of nodal and paranodal specializations. These features provide a core for further studies to determine: axon–myelinating cell communication; the structures of the proteins and lipids upon which myelinated fibers are formed; the pathways used to transport these molecules to sites of myelin assembly and maintenance; and the gene regulatory networks that control their expressions.

## 1. Introduction

Morphological, biochemical, cellular and molecular biological studies from species representing each vertebrate class have clearly shown that myelin structures in vertebrate central (CNS) and peripheral (PNS) nervous systems are highly conserved. These pleisomorphic traits may underlie the recent focus of myelin research for two species—mice and zebrafish [1,2,3,4]. Additional reasons for the growing domination of mice and zebrafish include the widespread availability and cost of these animals, facilities to house them and assess their behaviors, the wealth of mutants, ways to control and modify gene expression, and, for zebrafish, body transparency and the ability to watch myelination live. 

We propose expanding myelin research to additional species, e.g., species that will impact our understanding from different viewpoints including evolution, involvement in circuit wiring, the pathways underlying axon–myelinating cell communication, and adaptability following injury or disease. Sequences of many non-mammalian genomes, including elasmobranchs [5,6,7,8], gar [9], coelacanth [10], lamprey [11,12] and hagfish [13], coupled with the absence of an extra genome duplication characteristic of teleost fish and amphibians offer distinct possibilities. Additionally, slow evolutionary rates in many species [9,14,15,16] coupled with expanding uses in developmental and molecular biology [17,18,19,20,21,22,23,24,25,26], taken as a whole, make expansions to new species attractive. Undoubtedly, accompanying additions of new animal models will emerge techniques and approaches that will strengthen and re-enforce these choices and provide opportunities to test these concepts. 

We believe that one potentially useful species is the little skate, *Leucoraja erinacea*. Although not yet used for directed myelin research, another elasmobranch, the spiny dogfish, *Squalus acanthias*, has been used to show high similarities in morphological features present in the central (CNS) and peripheral (PNS) nervous system myelination with those of myelination in rats [27,28,29]. Appearances of myelinating oligodendrocytes and Schwann cells were included, as were the structural and ultrastructural features of internodes, nodes and paranodes. Studies with spiny dogfish embryos revealed that oligodendrocytes originate in comparable locations as they do in rat spinal cord and brainstem, and they ensheath and wrap axons indistinguishably from rat oligodendrocytes [27,29,30,31].

Specific advantages of using little skate over spiny dogfish are: 1) ease of obtaining embryos; 2) external development in skate egg cases (oviparity) [32]; and 3) a research history of widespread uses in neurophysiology [25,33], neurobiology [26] and developmental biology [18,32,34,35]. With the partially complete little skate genome [5], plus full genomes available from other elasmobranchs [6,7,8,36,37] coupled with comprehensive transcriptomes from spiny dogfish [38] and ocellate spot skate [39], information for generating reagents to use in studying oligodendrocyte and/or Schwann cell development in situ using these species is plentiful. Additionally, some antibodies, raised to mammalian proteins, recognize elasmobranch homologs [26,30,40].

As a basis for future studies, we examined structural and ultrastructural features in the CNS (spinal cord and optic nerve) and PNS (ventral root) of a little skate hatchling following the parameters used for mammalian and zebrafish myelination studies. Our findings are presented in this report and include low and high magnification EM images, g-ratio calculations from small, medium and large caliber axons, and structural examples of myelin outfoldings and Schmidt–Lanterman incisures (SLIs).

## 2. Materials and Methods

### 2.1. Tissue Preparation

A young skate, *Leucoraja erinacea* (7.5 cm in length, Figure 1A), obtained from the Marine Resources Center, Marine Biological Laboratory, was euthanized by anesthetic overdose (aqueous, buffered solution of Tricaine methane–sulfonate (MS-222; 100 mg/L seawater) following an approved Institutional Animal Care and Use Committee (IACUC) protocol (Marine Biological Laboratory). When the animal was insensitive to touch/pinching, it was pinned ventral side up, the heart was exposed and fixative (3% glutaraldehyde, 1.5% paraformaldehyde, in 0.1 M cacodylate buffer containing 10 mM MgCl_2_ and EGTA. Tonicity was at 1100 mOsM with 0.3 M sucrose) was administered first by cardiac perfusion, and then by direct application following opening the cranial and spinal cavities. The brain with attached optic nerves and proximal spinal column were removed (Figure 1B), transferred to fresh fixative, and stored overnight at 4 °C. The spinal cord and optic nerves were separated from the brain, and the fixative was replaced with cacodylate buffered sucrose (1100 mOsM), packaged and sent by FedEx to Germany.

### 2.2. Light and Electron Microscopy

Little skate spinal cord and optic nerves were embedded for observation in the transmission electron microscope (TEM) according to [41] and for large tile scans in the scanning electron microscope (SEM) using a rOTO (reduced OsO_4_/thiocarbohydrazide/OsO_4_) protocol [42]. Plastic embedded blocks were sectioned for semi- and ultrathin sectioning with a 35° diamond knife (Diatome, Biel, Switzerland) and a UC7 ultramicrotome (Leica Microsystems, Vienna, Austria). Ultrathin sections were transferred to TEM grids (100 mesh hexagonal copper grids, Science Services, Munich). The rOTO-embedded samples were cut using an array tomography S ATS diamond knife (Diatome, Biel, Switzerland) and semithin sections were placed on 10 mm × 10 mm silicon wafer (Science Services, Munich, Germany) which was 5 nm carbon coated with a Leica ACE600 Sputter coater (Leica Microsystems, Vienna, Austria). Ultrathin sections were imaged with a Zeiss LEO912 TEM (Carl Zeiss Microscopy, Oberkochen, Germany). Semithin sections on a wafer were sputter coated again with 5 nm carbon. Large tile scans were performed in a Zeiss Crossbeam 540 focused ion beam scanning electron microscope using ATLAS5 (Mosaic tile scan, Fibics/Carl Zeiss Microscopy, Oberkochen, Germany). The images were acquired with a back scattering detector BSD1 at 3 kV at a pixel size of 15 nm and a dwell time of 12 µs. For the light microscopic observation, semithin sections of samples prepared for TEM analysis were placed onto objectives slides, stained with Methylene blue/Azur II and imaged with a Zeiss AxioImager Z1 equipped with a Zeiss AxioCam MRc camera at 40× magnification using the ZEN software (Carl Zeiss Microscopy, Jena, Germany).

### 2.3. G-ratio Measurements

For measuring myelin thickness, g-ratios were determined from SEM-tile scans using Fiji (https://imagej.net/Fiji). Axon and corresponding myelinated fiber diameters were calculated from the measured area (A) using the equation (Equation 1):(1)$$d=2\sqrt{\frac{A}{\pi }}$$

The g-ratio was obtained by dividing the axonal caliber by the total fiber diameter. In fibers with an enlarged inner tongue, the inner myelin diameter was used for g-ratio calculations and the g-ratio was plotted in relation to axon caliber.

For the light microscopic observation, semithin sections of samples prepared for TEM analysis were placed onto objectives slides, stained with Methylene blue/Azur II stain and imaged using a Zeiss AxioImager Z1, equipped with a Zeiss AxioCam MRc camera at 40× magnification using the ZEN software (Carl Zeiss Microscopy, Jena, Germany). 

## 3. Results

### 3.1. Little Skate Hatchling Spinal Cord

The little skate hatchling spinal cord (Figure 2) seen in the transverse section was organized like spinal cords of other jawed vertebrates [26,43,44] including spiny dogfish and rat (Figure 3 [27]). Myelinated fibers concentrate in the white matter (WM) that surrounds sparsely myelinated gray matter (GM). The largest (myelinated) fibers develop in the ventral (VR) and dorsal (DR) roots and the medial fascicle of Stieda (FMS, fiber bundles specific to elasmobranchs [43,45]). Relatively large myelinated fibers populate the ventral funiculi (VF). At this developmental stage, the majority of myelinated fibers in the lateral (LF) the dorsal (DF) funiculi are small in caliber. Along with the FMS bundles are two more symmetrically arranged myelinated fiber bundles located in the dorsal GM (Figure 2, arrowheads). The circuitry of these fiber bundles is not known.

To assess myelination in the different areas of the spinal cord in their context, a cross section equivalent to Figure 2 was obtained by scanning electron microscopy (SEM) with detection of backscatter electrons. Overlapping large image tiles were stitched to generate a complete cross-section of the spinal cord with 17 nm pixel size. SEM images six areas of little skate spinal cord (Figure 2, A1–A6), used to determine g-ratios (4.3), highlight regional variations in myelinated fiber sizes as well as the relationships of myelin sheath thicknesses to axon calibers (Figure 3 and Figure 4).

All the fibers in the spinal cord sensory regions that include the dorsal funiculus (Figure 3A/A1), upper medial fiber bundles (Figure 3B/A2) and lateral funiculi (Figure 3C/A3) are of small caliber and thinly myelinated, presumably due to a combination of size limitations and/or slow development. In contrast, a significant portion of myelinated fibers in motor regions, i.e., the ventral funiculi (Figure 3D/A4 and Figure 3E/A5) and the medial fascicle of Stieda (Figure 3F/A6) are of large caliber. High numbers of oligodendrocyte somata positioned among myelinated fiber clusters reflect active myelination (Figure 3, “O”). Although most myelinated fibers in each region display typical ‘donut-shape’ appearances, a reasonable number of fibers not directly contacting oligodendrocyte somata display myelin outfoldings. Distributions of these myelin outfoldings vary greatly. One example is the difference between a single myelin outfolding in the area of dorsal funiculus analyzed for g-ratios (Figure 3A/A1, arrows) and the contralateral area, which contains six myelin outfoldings (Figure 3A/A1, inset). In general, fibers with myelin outfoldings occur throughout the cord, including in all the areas analyzed for g-ratios (Figure 3A–G). Schmidt–Lanterman incisures (SLIs), cytoplasmic channels seen as openings between regions of compacted myelin, are rare. None were seen in the small piece of ventral root examined (see below) and only a few were found in the sections of spinal cord and optic nerve examined; one and possibly two appeared in A4 (Figure 3G), with the clearest example occurring in the ventral funiculus (Figure 3H). We also observed membrane accumulations of varying sizes and structures in many the of large caliber axons in the ventral funiculus and most notably in the medial fascicle of Stieda (see Figure 3E, *). With the exceptions of myelin outfoldings and axon membrane accumulations present in many large caliber axons, the perusal of skate spinal cord sections show that they are structurally very similar to those of other gnathostome (mammals, birds, reptiles, amphibians and teleost fish) spinal cords.

### 3.2. Myelin Thickness in Fibers in Little Skate Hatchling Spinal Cord

Clearly, as in all myelinated nervous systems, myelin sheaths of different thicknesses cover axons of different calibers. These differences are frequently quantified with g-ratios (inner myelin sheath diameter/outer myelin sheath diameter [46,47]). We measured g-ratios in six areas (Figure 2, A1–A6) that cover the entire gamut of central axon calibers present in the spinal cord, and generated g-ratio versus axon caliber plots for each region (Figure 4). 

Although the smallest myelinated axons counted were <2 µm, smaller myelinated axons were also present in lower numbers, but were missed in the analysis. Small caliber myelinated fibers are frequent in the optic nerve (4.4).

Variations in the distributions in myelinated axon calibers among the different areas are quantified here. In the dorsal funiculus (A1), most axons are of small caliber, 2.5–6 µm, with g-ratios more variable among the small caliber axons. In fiber bundles in the dorsal horn fiber bundle (A2), axon calibers cover a wider range (2.5–10 µm), with slightly greater variability in the g-ratios of smaller caliber axons. In a region of lateral funiculus (A3), axons are mostly of small caliber (2.5–8 µm) with few larger than any in dorsal funiculus or the dorsal horn fiber bundle areas examined. In the upper ventral funiculi (A4), the majority of axons are of small caliber (3–8 µm), though many medium and larger caliber axons are present. Clearly, the axons of largest caliber reside in the lower ventral funiculi (A5) and especially in the medial fascicle of Stieda (A6). Although as in other areas, portions of small caliber axons (2.5–10 µm) are large, substantial numbers of large caliber (10–20 µm, A5 and 10–25 µm, A6) are present. G-ratios in all regions examined hover around 0.7, a value common among g-ratios of axons in mammalian CNS [46]. These results demonstrate that developmental mechanisms regulating axon-dependent myelin sheath expansions are conserved in little skate.

### 3.3. Ultrastructural Features of Little Skate Spinal Cord

Whereas tetrapods, including mammals, use proteolipid protein (PLP) to compact extracellular domains during active myelination, cartilaginous and teleost fish use myelin protein zero (MPZ) [48,49,50,51]. Although these differences result in differences in thicknesses of interperiod lines (where extracellular surfaces compact) [52], they do not appear to modify axon–myelin sheath relationships present among fish and tetrapods [27,28,29,31]. Ultrastructural features of myelination have not been examined in little skate, and active stages of myelination have been omitted in studies using spiny dogfish and bamboo sharks [27,29,30,31]. Using the little skate hatchling, we assessed the ultrastructural features of myelinating oligodendrocytes in spinal cord (Figure 5) and optic nerve (Figure 6). We also examined Schwann cell myelination occurring in developing ventral root (Figure 7).

As noted above, myelinating oligodendrocytes are far more abundant in little skate hatchling spinal cord white matter (Figure 3) compared with the adult (not shown). Three myelinating oligodendrocytes are shown at high magnification (Figure 5A–C), one from the dorsal funiculus (A), and two from the lateral funiculus (B, C). Axons of varying calibers are associated with each oligodendrocyte soma and none have myelin outfoldings. As demonstrated from g-ratio measurements, the thicknesses of myelin sheaths surrounding each axon are of an appropriate size for an axon caliber. Two (Figure 5B,C) are sectioned at the level of oligodendrocyte nuclei, which have heterochromatin distributions, i.e., lining the nuclear envelop and clumped more centrally, similar to mammalian oligodendrocytes [53]. The higher magnification images (Figure 5C and insets) show that the cytoplasm is rich in stacked rough endoplasmic reticulum (rER), mitochondria, lysosomes, and Golgi apparatus (not shown). Dispositions and appearances of inner and outer tongue processes are marked (Figure 5C, lower inset) and appear indistinguishable from those of mammalian oligodendrocytes. Although axon membranous inclusions appear to be mostly associated with large caliber axons, three axons of relatively small caliber display distinct myelin inclusions at this magnification (Figure 5C, *), thus indicating they are quite common. Whereas nodal and paranodal specializations are difficult to visualize in transverse sections, they are common in longitudinal sections. One example shows profiles of paranodal loops approaching the bare node from each side. (Figure 5D). Here again, these structures appear indistinguishable from the same specializations as described for mammalian nodes [54].

### 3.4. Structural and Ultrastructural Properties of Little Skate Optic Nerve

An entity common among myelination studies in all jawed vertebrates is the optic nerve. At the time of hatching, the little skate optic nerve is replete with myelin fibers of varying calibers (Figure 6A). As a consequence of its late development, the optic nerve contains an abundance of unmyelinated axons, and those that are myelinated are no larger than the largest caliber axons in the dorsal funiculi (Figure 2 and Figure 3A). Higher magnification images (Figure 6B–D) show that as in a developing spinal cord, myelinating oligodendrocyte somata replete with complements of myelinating fibers are abundant in the optic nerve. As with spinal cord axons, several of the optic nerve myelinated axons contain myelin outfoldings (Figure 6B–D, arrows). None are as lengthy or abundant as in the spinal cord. Few Schmidt–Lanterman incisures are seen. We illustrate one in Figure 6B (“S”).

To be complete, we include a high magnification TEM images of areas containing many small caliber unmyelinated fibers with few myelinated fibers and associated oligodendrocyte processes (Figure 6E,F). We show one unusually large myelin outfolding that extends outward from the axon being myelinated along a thin oligodendrocyte/astrocyte process and many unmyelinated axons and ending in a loop (Figure 6F). 

As we did with the spinal cord, we analyzed g-ratios for myelinated fibers in an optic nerve on SEM tile scans (Figure 6G). All axons were of small caliber (0.5–2.5 µm) and the regression line was lower than those of the larger axons analyzed in the six spinal cord areas.

### 3.5. Structural Features of a Ventral Root

Additionally, we examined a single section of ventral root extending from the little skate hatchling spinal cord (Figure 7). Most myelinated fibers in the ventral root are larger than those in the overlying ventral funiculus. Although the large axons in the intermediate and more distal regions of the root are myelinated by Schwann cells (many profiles with perinuclear Schwann cell cytoplasm are seen), those in the more proximal transition zone (Figure 7, TZ) are myelinated by oligodendrocytes. Although the EM section is from an elasmobranch utilizing myelin protein zero-based myelin fibers, we do not see any Schmidt–Lanterman incisures. However, a few myelin outfoldings are seen (Figure 7A, arrows). Higher magnification images show several profiles cut of a region with perinuclear Schwann cell cytoplasm (Figure 7B, “SC”). The Schwann cell displays a nucleus containing clumped heterochromatin along the inner edge with a single small fiber that has been separated from neighbors but not yet myelinated. In the cross-section, figures with nodes and paranodal loops are rare. The one found is shown in Figure 7D. Overall, as seen in the CNS and in previous studies that examined PNS myelination in spiny dogfish [27] and Atlantic stingray [28], the structural features appear indistinguishable from those of mammals and other vertebrate classes.

## 4. Discussion

In recent decades, research to uncover mechanisms of myelination have increasingly relied on experimentation from two principal animal models: mice and zebrafish. Likely, this trend will continue since studies of myelination conducted by the majority of labs, principal investigators, students and postdoctoral fellows, use these animals. Why then, should we raise the possibility of adding more species? Certainly, many species have been used throughout the history of myelination studies and selective narrowing has results from the development and uses of gene manipulation approaches designed for mice and zebrafish. We argue that it is time to expand myelin research to new, less frequently used species. In this way we can learn how widespread the properties being uncovered are, and whether features identified in mouse and/or zebrafish studies can be traced to a common ancestor or arose later. Although some comparative studies between mice and zebrafish are being used to address this issue [55,56,57], how can one follow up those studies with features that differ [4]? Obviously, the availability of additional species will be helpful in sorting out alternative mechanisms when they are uncovered.

Advantages of zebrafish including their small size, rapid growth, achievability of large-scale screening to identify and characterize new ‘myelination’ genes and pathways, and the ability to observe myelination in vivo [1,4,58] are indeed attractive. However, one must be concerned that biases will arise focused on a single small rapidly developing nervous system.

Recently, Zalc and colleagues have taken advantage of their size and body transparency to introduce transgenic *Xenopus laevis* tadpoles as a model that will aid in identifying small molecules that promote remyelination [59,60,61]. Hopefully, these studies will not be limited to this purpose alone but will be expanded to tackle questions on fundamental myelination mechanisms.

The self-evident advantages of *Xenopus* tadpoles and zebrafish attributable to their small sizes, body transparencies and rapid development must be balanced by complexities introduced by genome duplication and allotetraploidy [62,63,64]. Added genome duplication results in additional copies of many proteins with consequential effects on gene dosage, regulatory control, and protein–protein interactions.

Elasmobranch, gar and coelacanth genomes, unlike zebrafish and *Xenopus* tadpoles, have the advantages of not undergoing additional chromosomal duplications and as well as possessing a more slowly evolving evolutionary history [14,15,16,64]. Although the little skate, *Leucoraja erinacea*, is the focus of this study, an alternative, complementary species, the brown banded bamboo shark, *Chiloscyllium punctatus*, has similar advantages, genome data and the accessibilities of embryos [65] as well as a preliminary study on myelination [31]. We contend that many other species with myelinated nervous systems will have features appealing for a variety of studies.

Both the little skate and bamboo shark have nervous systems with a size and complexity similar to those of rodents. Both species are oviparous, egg-laying, direct developers that allow easy access to embryos throughout development [32,65,66]. However, their body plans and behaviors are very different, i.e., the bamboo shark locomotes like other fish whereas little skates walk and use gigantic pectoral fins for a more benthic lifestyle, thus offering studies comparative in nature.

Together, earlier morphological studies on myelination in the brain, spinal cord and trigeminal nerves of spiny dogfish, *Squalus acanthias* [27,29,30,67], brown banded bamboo shark, *Chiloscyllium punctatus* [31], and the dorsal and ventral roots of the Atlantic stingray *Dasyatis sabina* [28] demonstrate that the structural and ultrastructural features of early and mature stages of myelination are virtually indistinguishable from features in the brains, spinal cords and peripheral nerves of mammals and species from other vertebrate classes [68,69]. Features that appear to be homologous include the appearances of myelinating cells, oligodendrocytes and Schwann cells, and the structure and organizations of myelin internodes, paranodes and nodes of Ranvier.

In this paper, we not only examined for the first-time myelination in little skate, but we chose a stage with widespread active myelination not previously examined. Compared with the spinal cords and optic nerves of more mature elasmobranchs and mammals [27], the spinal cord and optic nerves of developing little skate contain far higher densities of myelinating oligodendrocytes, particularly in regions replete with small caliber fibers (Figure 3 and Figure 6). Many axons undergoing active myelination are in direct contact with oligodendrocyte somata. Notably few, if any of them, exhibit myelin outfoldings. Myelin outfoldings appear to be associated with axons separated from oligodendrocyte somata, suggesting they reside in the internodal regions distant from the somata. Structural appearances of many of the larger myelin outfoldings indicate that they fold back over myelin sheaths (Figure 3 and Figure 6). Far less frequently do we see outfolding radiating away from myelinating axons (Figure 6F) suggesting some limitations to radial growth. 

Myelin outfoldings were first described in some detail in fibers found in an adult toad cerebellum [70]. They are rarely observed in healthy adult mammalian CNS nor in spinal cords of mature spiny dogfish [27] and little skate (unpublished observations). They are far more commonly associated with the myelin sheaths of mice formed under compromised conditions, i.e., the absence of myelin-associated glycoprotein [71], the absence of 2’,3’-cyclic nucleotide 3’-phosphodiesterase [72], the absence of PLP [73] and/or diminished anillin/septin scaffolds [73,74], myotubularin 2-related protein [75], as well as actin bundling-related structures [76]. One possible explanation for their presences in developing little skate spinal cord, optic and spinal nerves is that the myelin outfoldings, apparently more frequent and larger in smaller fibers, are due to an abundant production in outlying portions of internodes that are used to accommodate myelin membrane expansion around the axons as they enlarge in caliber and length. Under compromised conditions, the persistence of myelin outfoldings may reflect miscommunication between axons and the overlying myelinating cell processes supplying myelin constituents.

Ultrastructural features of myelinating oligodendrocytes and myelinated fibers in skate spinal cord (Figure 4) and optic nerve (Figure 6) include the clumped heterochromatin in oligodendrocyte nuclei, abundant rER and Golgi (not shown) in perinuclear cytoplasm, the close associations of adjacent myelin sheaths with one another, as well as distinct inner and outer tongue processes, nodal and paranodal specializations. These features appear indistinguishable from those described for nervous systems of mammals and other vertebrate classes [53,69,77]. Similarly, the Schwann cell-based PNS (Figure 7) display features that appear indistinguishable from those of mammals and other vertebrate classes [53].

Although SLIs are common features of frog, mouse and human peripheral nerves [78,79,80], they are rare in the central nervous systems [69,81]. Finding very few in developing skate CNS and PNS (Figure 3, Figure 6 and Figure 7) and in young adult spiny dogfish [27] suggests that their formation does not correlate with myelin protein zero (MPZ) levels, as suggested in a study in which MPZ was introduced as a replacement for PLP in mouse brain [82]. These results show that their presence and need are of limited importance in myelin sheaths that have MPZ as the dominant protein.

Finally, we see numerous membranous inclusions in large caliber axons. These inclusions may represent material in transit, either anterogradely to terminals or retrogradely back to soma. They might also represent excess myelin membranes endocytosed prior to their removal. Immunohistochemical studies to identify the proteins present in these structures should help explain their presence.

What potential uses could little skate, bamboo shark, and other animals have in advancing myelination studies? With many genomic sequences now available, coupled with ample amounts of tissues (brain, spinal cord and peripheral nerves), a number of species could provide isolated myelin for proteome, lipidome and transcriptome studies. Proteome study will require the establishment of a reference database for mass spectroscopy. All these approaches have been used on numerous occasions with myelin prepared from rodents, and in some cases, humans [83,84,85,86,87,88,89,90,91,92,93,94,95,96,97], providing a plethora of data for comparative studies.

Since it is possible to culture elasmobranch oligodendrocyte lineage cells and these cells express surface antigens recognized by mammalian antibodies [30], these cells could be isolated and used for transcriptome studies similar to the approaches used to generate a widely used database of mouse oligodendrocyte lineage cells and other neural cell subtypes [98]. It should also be possible to dissect single oligodendrocyte lineage cells directly from tissues as was done for mouse and zebrafish [99,100,101,102,103], generating starting material for single cell transcriptome studies, studies that would determine which mRNAs are common to oligodendrocyte lineage cells in mice and little skate. Follow-up in situ hybridization and immunohistochemical studies would then be conducted to determine whether the locations and timings of mRNA, protein and lipid expressions correspond to those occurring during mammalian myelination. These efforts would apply to molecules in compact myelin, in adaxonal, abaxonal and in paranodal regions.

Elasmobranchs are situated at a critical juncture in vertebrate evolution. Studying the evolution of myelin using one or more elasmobranchs will hopefully impact our knowledge of the role of myelination in neuronal circuitry development, in signaling between neurons/axons and myelinating cells, structural features of the proteins and lipids used to form and maintain myelin sheaths and the developmental behaviors and gene expressions underlying the recruitment and differentiation of oligodendrocyte progenitors in areas where myelination is occurring.

## Figures and Tables

**Figure 1 life-11-00136-f001:**
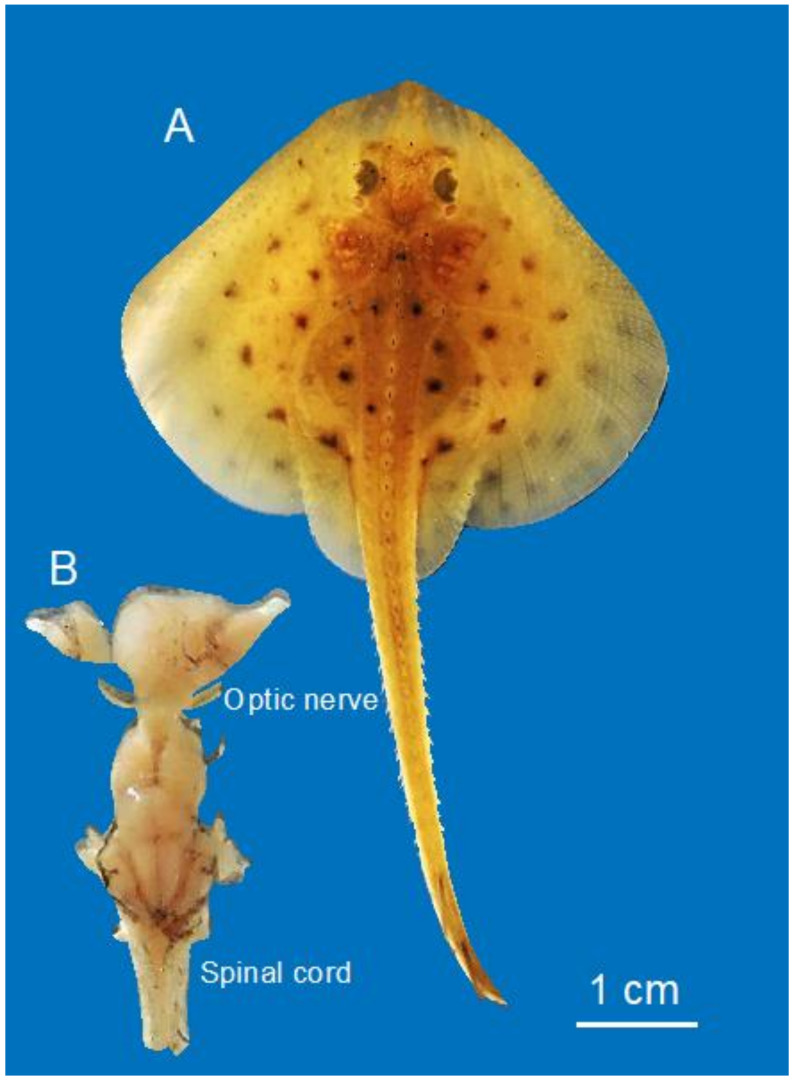
Little skate hatchling (**A**) and dissected brain (**B**) used in this study.

**Figure 2 life-11-00136-f002:**
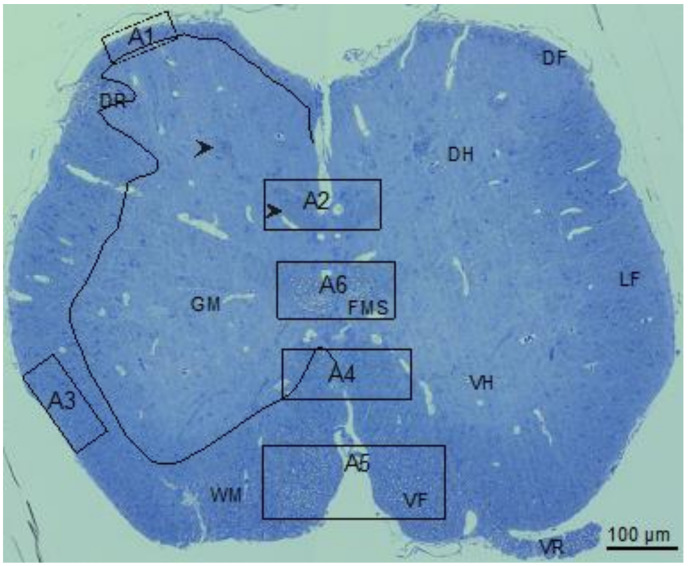
Little skate hatchling spinal cord. Left side shows regions of gray (GM) and white (WM) matter separated by a multi-curved line. An entering dorsal root (DR) is seen in this hemisection. Right side includes marking for the dorsal funiculus (DF), dorsal horn (DH), medial fascicle of Stieda (FMS), lateral funiculus (LF), ventral horn (VH), ventral funiculus (VF) and ventral root (VR). Areas containing small, medium, and large fibers used for the calculation of g-ratios are marked (A1–A6). On the left hemisection, arrowheads show two myelinated fiber bundles within the DH, the lower one used for g-ratio determination. Bar shows magnification.

**Figure 3 life-11-00136-f003:**
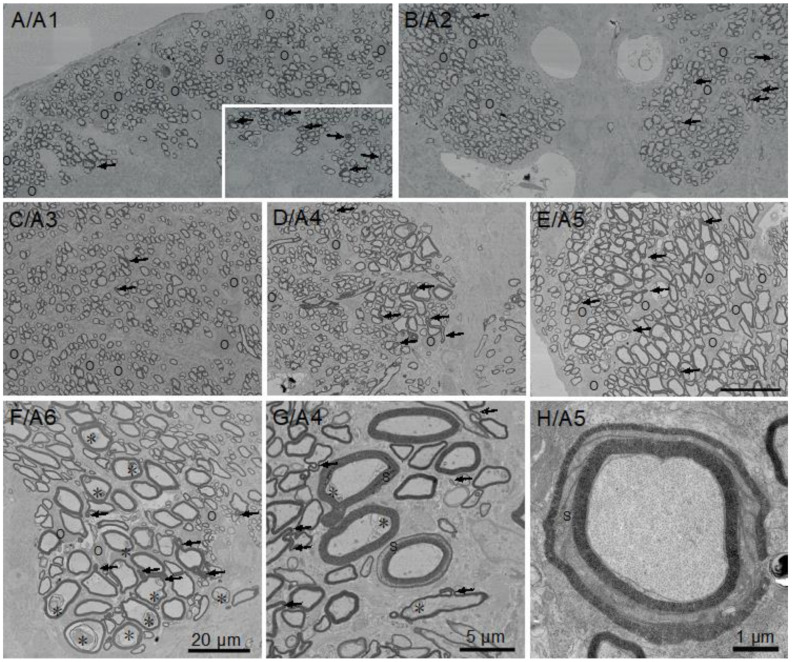
Little skate hatchling spinal cord imaged by scanning electron microscopy (SEM) showing the six areas (A1–A6) used to measure g-ratios (Figure 4). The inset in A/A1 illustrates the differences in numbers of myelin outfoldings in area analyzed (A1, one) versus a smaller region on the contralateral side (six). Myelin outfoldings in other regions of the spinal cord are shown (arrows), as are membrane inclusions in axons (F/A6 and G/A4). Images A/A1 to F/A6 are at the same magnification (20 µm). G and H are at higher magnification, with H obtained by transmission electron microscopy (TEM). Symbols: O: oligodendrocyte soma, →: myelin outfoldings, S: Schmidt–Lanterman incisure, *: membrane inclusions in large axons.

**Figure 4 life-11-00136-f004:**
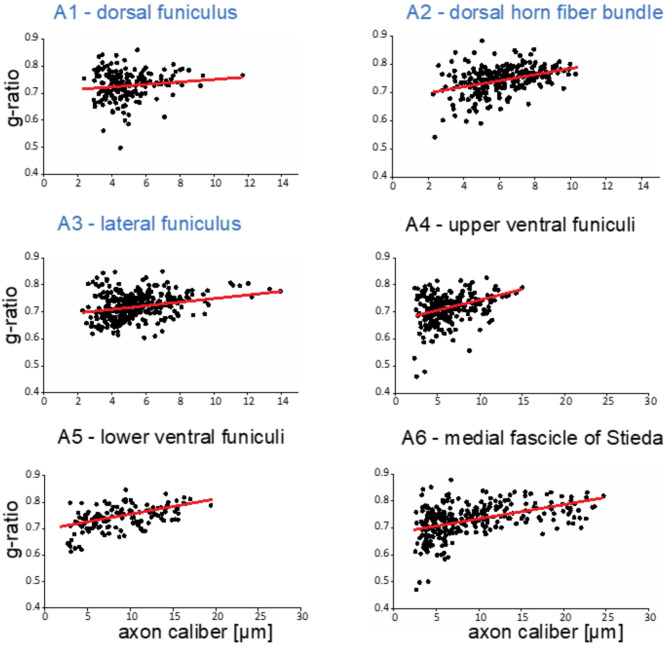
G-ratios of myelinated fibers in different regions of the little skate spinal cord. G-ratios were measured (methods) in six areas of the developing spinal cord (Figure 2, A1–A6). Maximal axon calibers in A1–A3 are different from those in A4–A6 as noted in the different coloring of the figure subtitles. Regression lines are shown in red and intersect near 0.7.

**Figure 5 life-11-00136-f005:**
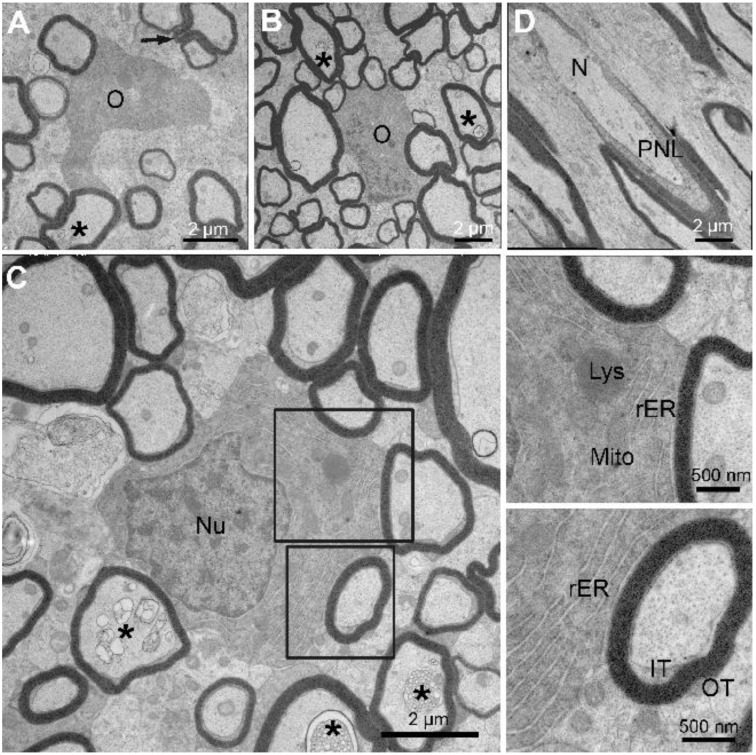
Little skate hatchling spinal cord—the structural features of myelinating oligodendrocytes and a node of Ranvier. TEM images showing three myelinating oligodendrocytes are shown (**A**–**C**). Each is associated with axon undergoing myelination with sheath thicknesses reflecting the axon caliber. Two enlargements of an organelle-rich cytoplasm from Figure 4C. (**D**) Longitudinal image sectioned at a node of Ranvier. Abbreviations: inner (IT) and outer (OT) tongues; node of Ranvier (N); bordering paranodal loops (PNL); O, oligodendrocyte; Nu, oligodendrocyte nucleus; Lys, lysosome; rER, rough endoplasmic reticulum; Mito, mitochondria; * membrane inclusions in axons.

**Figure 6 life-11-00136-f006:**
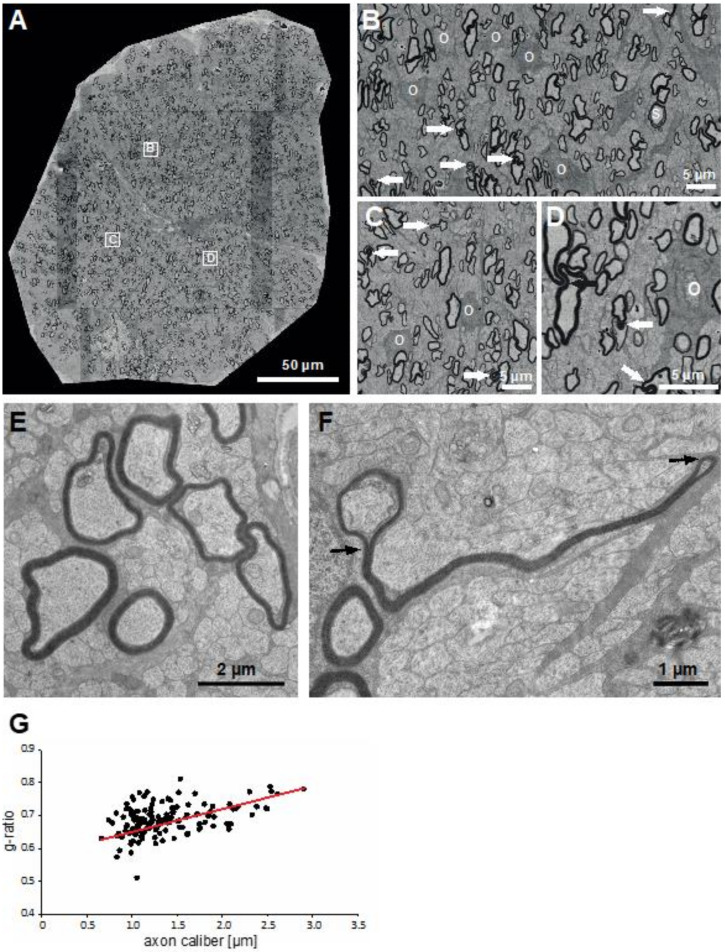
Little skate hatchling optic nerve. Whole optic nerve imaged by SEM (**A**), enlargements of three areas (**B**–**D**), ultrastructural images obtained by TEM (**E**,**F**) and g-ratio analysis (**G**). Abbreviations: O: myelinating oligodendrocytes; →: myelin outfoldings.

**Figure 7 life-11-00136-f007:**
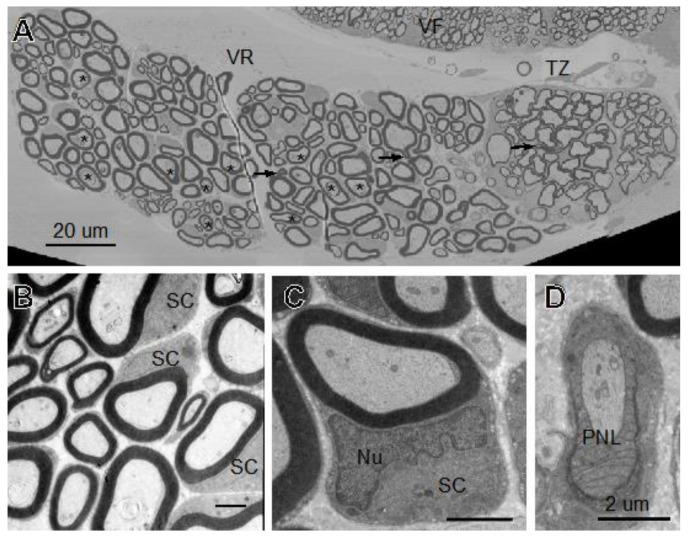
Little skate hatchling ventral root shown at low magnification in SEM (**A**). Fibers are clearly larger than those in the overlying spinal cord. (**B**,**C**) Show higher magnification images of myelinated fibers including some surrounded by perinuclear cytoplasm. (**D**) Fiber showing paranodal loops (PNL). Like D, magnification bars in B and C are 2 µm. Abbreviations: VR: ventral root, VF: ventral funiculus, TZ: transition, SC: Schwann cell, Nu: Schwann cell nucleus, PNL: paranodal loops, →: myelin outfoldings, *: membrane inclusions in large axons.

## Data Availability

Data is contained within the article or supplementary material.

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
