# Peer review of "New Species Can Broaden Myelin Research: Suitability of Little Skate, *Leucoraja erinacea"

_life, 2021, doi:10.3390/life11020136_

Round 1
Reviewer 1 Report
The manuscript entitled“Myelin studies need new models: suitability of little skate, Leucoraja erinacea” is focusing on the value of the little skate as a novel animal model in the research field of myelin. In this research, the authors presented basic morphological features at spinal cord and optic nerve in little skate hatchling. It is very interesting that the authors discuss differences between little skate and mice, which are the most common animal models for myelin research. However, the figures and text need to be improved in several points for publish.
The authors should address the following points:
Comment 1: Many people who will read this paper will not know about little skate. Representative pictures of the little skate hatchling used in this study should be added. Pictures of “brain and spinal cord specimens" should be inserted together if you can.
Comment 2: It is confusing that the presence of Schmidt-Lanterman incisures (SLI) in the spinal cord. SLI is a specific feature of myelinated axons formed in Schwann cells in the peripheral nervous system (PNS). The authors mention that LSI was not detected in the PNS, although a small number of LSI was observed in the spinal cord and optic nerve of little skate. If the absence of LSI was due to immatureness of the PNS as described in discussion, then why was LSI observed in the spinal cord of the little skate? Do these results suggest that little skate and spiny dogfish have a specific component of myelin proteins in oligodendrocytes as described in discussion? The authors should explain their speculations carefully and clearly.
Comment 3: The g-ratio should also be measured in the optic nerve and peripheral nerves (Figure 5 and 7) if they mention to differences in axon-myelin relationships among these organs as described in results section.
Minor points
1) In Figure 3, the x-axis, y-axis, and graphical size of all graphs should be adjusted with area 5 graph.
2) I think “SEM” is not correct. “TEM” is right?
3) Insert period at lane 206.
4) Insert (D) at lane 211
Author Response
The manuscript entitled“ Myelin studies need new models: suitability of little skate, Leucoraja erinacea” is focusing on the value of the little skate as a novel animal model in the research field of myelin. In this research, the authors presented basic morphological features at spinal cord and optic nerve in little skate hatchling. It is very interesting that the authors discuss differences between little skate and mice, which are the most common animal models for myelin research. However, the figures and text need to be improved in several points for publish.
The authors should address the following points:
Comment 1: Many people who will read this paper will not know about little skate. Representative pictures of the little skate hatchling used in this study should be added. Pictures of “brain and spinal cord specimens" should be inserted together if you can.
I agree that pictures of little skate hatchling and brain and spinal cord will be helpful. These were added in a new Figure 1 and are referred to in the Methods section (lines 93 and 103).
Comment 2: It is confusing that the presence of Schmidt-Lanterman incisures (SLI) in the spinal cord. SLI is a specific feature of myelinated axons formed in Schwann cells in the peripheral nervous system (PNS). The authors mention that LSI was not detected in the PNS, although a small number of LSI was observed in the spinal cord and optic nerve of little skate. If the absence of LSI was due to immatureness of the PNS as described in discussion, then why was LSI observed in the spinal cord of the little skate? Do these results suggest that little skate and spiny dogfish have a specific component of myelin proteins in oligodendrocytes as described in discussion? The authors should explain their speculations carefully and clearly.
I totally agree that the test would have been confusing to the reviewer. It stems from our bias, amplified in the reference of Yin 2008, that correlates MPZ with Schmidt-Lanterman incisures. In fact, these studies which show the paucity of SLIs in both CNS and PNS tissues (line 197) suggests that the correlation of MPZ and SLI is probably not correct. This is the point we make in the discussion (paragraph beginning at line 417).
Comment 3: The g-ratio should also be measured in the optic nerve and peripheral nerves (Figure 5 and 7) if they mention to differences in axon-myelin relationships among these organs as described in results section.
We added an additional area of spinal cord (Figures 2 and 4, A5, lower ventral funiculi) as next to the medial longitudinal fascicle of Stieda, it contains largest myelinated fibers. We also explained that the range of sizes covered does not include axons <2 μm because they are missed/underrepresented using the grid-based method to select myelinated fibers to measure (line 221). Adding optic nerve (Figure 6G) allows us the opportunity to describe myelination of very small caliber axons and observe the sizes of smallest axons receiving myelin as well as a propensity for over myelination (g-ratios lower than those in spinal cord). We had too little material to analyze the spinal root.
Minor points
- In Figure 3, the x-axis, y-axis, and graphical size of all graphs should be adjusted with area 5 graph.
We made the change so now all six areas measured are with the same y-axis and one of two x-axes.
- I think “SEM” is not correct. “TEM” is right?
In fact, SEM is correct and was used to obtain high power images of entire spinal cord and optic nerve. We modified the materials and methods section to describe what we did.
- Insert period at lane 206.
We made this change.
- Insert (D) at lane 211
We made this change.

Reviewer 2 Report
- In the Intro, do the authors really mean that vertebrate myelinated CNS and PNS are highly conserved, or that myelin structure in the CNS and PNS is highly conserved across vertebrate species? I wouldn’t think that e.g. chicken brain and human brain can be described as highly conserved organs, although they certainly have conserved features.
- p2, “Immerge” should be emerge. “Identical” should be similar or comparable.
- General observation: little skate is highly unlikely to become a widely used model for myelination, if by “model” the authors mean an experimental surrogate for human myelination, which I think would be the commonly understood meaning in this context. The lack of naturally occurring mutants or transgenic methods will see to that, as well as the difficulty of obtaining skate/shark specimens or breeding them in a lab setting. For gaining clues about myelin evolution it could certainly be informative, and this could in turn lead to functional insights. I suggest toning down statements about the need for new and more myelin models per se. I accept that this is a matter of personal opinion. However, hoping that preliminary studies on little skate and banded bamboo shark will lead to further wide-ranging experimentation using these species (p12) is perhaps wishful.
- Fig. 2, the Schmidt-Lanterman incisure supposedly indicated by an “S” in Fig 2(A4) is not evident at this magnification/resolution. Can an improved image be shown? In Fig. 5, is it certain that the pale peri-axonal ring indicated is a Schmidt-Lanterman incisure, rather than simply an enlarged peri-axonal space? Schmidt-Lanterman incisures are considered to be a specialization of Schwann cell myelin in mammals. Are they also a specific feature of fish oligodendrocytes, related to MPZ expression? Some more explicit explanation should be added here or in the Discussion.
- p5, “most myelinated fibers … are small, ie at early stages of myelination.” Presumably the authors mean “myelin is thin”, as a small fiber diameter could reflect small axon diameter, not necessarily indicating an early stage of myelination?
- Bottom of p12, in addition to the three levels of inquiry into myelin evolution suggested here, could be added the evolution of myelin development. It would be interesting, for example, to identify oligodendrocyte precursors (OPCs) in elasmobranch species and compare their origins, developmental behaviour and gene expression to those of mammals and other fish.
Author Response
Reviewer 2
- In the Intro, do the authors really mean that vertebrate myelinated CNS and PNS are highly conserved, or that myelin structure in the CNS and PNS is highly conserved across vertebrate species? I wouldn’t think that e.g. chicken brain and human brain can be described as highly conserved organs, although they certainly have conserved features.
A good point and we rephrased the first sentence of the introduction as recommended by the reviewer (line 44).
- p2, “Immerge” should be emerge. “Identical” should be similar or comparable.
Good points – we changed immerge to emerge (line 61) and identical to comparable (line 71).
- General observation: little skate is highly unlikely to become a widely used model for myelination, if by “model” the authors mean an experimental surrogate for human myelination, which I think would be the commonly understood meaning in this context. The lack of naturally occurring mutants or transgenic methods will see to that, as well as the difficulty of obtaining skate/shark specimens or breeding them in a lab setting. For gaining clues about myelin evolution it could certainly be informative, and this could in turn lead to functional insights. I suggest toning down statements about the need for new and more myelin models per se. I accept that this is a matter of personal opinion. However, hoping that preliminary studies on little skate and banded bamboo shark will lead to further wide-ranging experimentation using these species (p12) is perhaps wishful.
This is a carefully written and thoughtful comment. We changed the title to reflect that we were not in fact introducing a new animal model, but simply proposing the usefulness of expanding myelin research to more and varied species, of which little skate is one possibility. At the end of the discussion, we suggest specific experiments that could provide novel information on properties of myelin, mRNA, protein and lipid constituents associated with little skate and myelinating cells, information that would be compared with similar data available from mice and other species to determine which molecules are shared and likely related to myelination originating in the common ancestor.
Fig. 2, the Schmidt-Lanterman incisure supposedly indicated by an “S” in Fig 2(A4) is not evident at this magnification/resolution. Can an improved image be shown? In Fig. 5, is it certain that the pale peri-axonal ring indicated is a Schmidt-Lanterman incisure, rather than simply an enlarged peri-axonal space? Schmidt-Lanterman incisures are considered to be a specialization of Schwann cell myelin in mammals. Are they also a specific feature of fish oligodendrocytes, related to MPZ expression? Some more explicit explanation should be added here or in the Discussion.
As stated in the comments to reviewer one, we rewrote the description of our SLI findings (paragraph starting at line 416), which in essence showed that choosing little skate, we add doubt to the notion that myelin sheaths with high MPZ content have high numbers of SLI. We add a high magnification of a SLI to Figure 3 (formerly Figure 2).
- p5, “most myelinated fibers … are small, ie at early stages of myelination.” Presumably the authors mean “myelin is thin”, as a small fiber diameter could reflect small axon diameter, not necessarily indicating an early stage of myelination?
We clarified the text stating the axon calibers were small and the myelin sheaths around them were thin
- Bottom of p12, in addition to the three levels of inquiry into myelin evolution suggested here, could be added the evolution of myelin development. It would be interesting, for example, to identify oligodendrocyte precursors (OPCs) in elasmobranch species and compare their origins, developmental behaviour and gene expression to those of mammals and other fish.
This was a thoughtful and important comment, and we included the notion in our concluding paragraph.
Round 2
Reviewer 1 Report
The authors responded to all my comments clearly.